# Opioids modulate an emergent rhythmogenic process to depress breathing

Xiaolu Sun[1†], Carolina Thörn Pérez[1†], Nagaraj Halemani D[2], Xuesi M Shao[1], Morgan Greenwood[3], Sarah Heath[1], Jack L Feldman[1], Kaiwen Kam[2]*

[1]Department of Neurobiology, David Geffen School of Medicine at UCLA, Los Angeles, United States; [2]Department of Cell Biology and Anatomy, Chicago Medical School, Rosalind Franklin University of Medicine and Science, North Chicago, United States; [3]RFUMS/DePaul Research Internship Program, Rosalind Franklin University of Medicine and Science, North Chicago, United States

**Abstract** How mammalian neural circuits generate rhythmic activity in motor behaviors, such as breathing, walking, and chewing, remains elusive. For breathing, rhythm generation is localized to a brainstem nucleus, the preBötzinger Complex (preBötC). Rhythmic preBötC population activity consists of strong inspiratory bursts, which drive motoneuronal activity, and weaker burstlets, which we hypothesize reflect an emergent rhythmogenic process. If burstlets underlie inspiratory rhythmogenesis, respiratory depressants, such as opioids, should reduce burstlet frequency. Indeed, in medullary slices from neonatal mice, the μ-opioid receptor (μOR) agonist DAMGO slowed burstlet generation. Genetic deletion of μORs in a glutamatergic preBötC subpopulation abolished opioid-mediated depression, and the neuropeptide Substance P, but not blockade of inhibitory synaptic transmission, reduced opioidergic effects. We conclude that inspiratory rhythmogenesis is an emergent process, modulated by opioids, that does not rely on strong bursts of activity associated with motor output. These findings also point to strategies for ameliorating opioid-induced depression of breathing.

*For correspondence:
kaiwen.kam@rosalindfranklin.edu

†These authors contributed equally to this work

Competing interests: The authors declare that no competing interests exist.

## Introduction

Rhythmic motor behaviors in mammals, such as breathing, walking, and chewing, are controlled by neural circuits that determine the period of the movement and shape the pattern of motor activity. The neural circuits controlling breathing are increasingly being recognized as relevant for elucidating mechanisms of rhythm and pattern generation and other principles of neural circuit function in mammals (*Grillner, 2006*; *Moore et al., 2014*; *Ruder and Arber, 2019*). Generation of breathing rhythm is localized to a compact nucleus in the ventrolateral medulla, the preBötzinger Complex (preBötC; *Smith et al., 1991*). Rhythmic bursts of preBötC population activity were previously considered unitary events that both determined the timing and initiated the pattern of periodic inspiratory movements. However, we showed that these preBötC bursts consist of two separable components: i) strong inspiratory bursts that are transmitted via premotoneurons to inspiratory motoneurons to activate muscles that produce inspiratory airflow; and ii) weaker burstlets that we postulated are an emergent property of the network (*Feldman and Kam, 2015*; *Kam et al., 2013a*). Burstlets appear as preinspiratory activity when preceding preBötC bursts and are not normally seen in motor nerve or muscle activity (*Kam et al., 2013a*).

Because preBötC rhythmic activity does not rely on intrinsic 'pacemaker' conductances or inhibitory synaptic transmission and can consist solely of burstlets that do not produce motoneuronal output, we hypothesize that inspiratory rhythmogenesis is an emergent process, distinct from burst

generation, that manifests as burstlets (*Feldman and Kam, 2015*; *Kam et al., 2013a*). In this model, burstlets occur when spontaneous firing among preBötC neurons reaches a threshold for synchrony, and the time required for preBötC neurons to assemble and achieve this synchrony (percolation) in each cycle is an essential determinant of the period of the rhythm (*Ashhad and Feldman, 2019*; *Feldman and Kam, 2015*; *Kam et al., 2013a*). A strong prediction of this hypothesis is that manipulations that slow breathing act specifically on this emergent rhythmogenic mechanism.

To test this prediction, we examined the effects of opioids, analgesics that significantly depress breathing at high doses, on bursts and burstlets and tested opioidergic interactions with other neurotransmitters and neuromodulators that affect preBötC neuronal excitability. We bath-applied [D-Ala$^2$,N-MePhe$^4$,Gly-ol$^5$]-enkephalin (DAMGO), a potent synthetic µ-opioid receptor (µOR) agonist, in conditions of low excitability in vitro when burstlets appeared at times bursts would have been expected or when preBötC rhythmic activity consisted solely of burstlets. We find that opioids depress inspiratory frequency by acting on burstlet-producing preBötC Dbx1-derived (Dbx1$^+$) neurons, an important glutamatergic subpopulation. These data are consistent with our hypothesis that an emergent mechanism, discernable as burstlets and independent of bursts, governs the cyclical assembly and activation of preBötC neurons underlying inspiratory rhythmogenesis. Our results also inform potential approaches for combating opioid-induced depression of breathing.

## Results

### DAMGO slows preBötC rhythmic activity without affecting burstlet fraction

Systemic administration of opioids slows breathing in rodents in vivo and in vitro (*Boom et al., 2012*; *Gray et al., 1999*). Inspiratory activity driving breathing can be assayed using extracellular field electrodes placed in the preBötC that measure action potentials generated by neurons near the electrode tip (*Kam et al., 2013a*; *Telgkamp and Ramirez, 1999*). This preBötC population activity is typically integrated, which accentuates activity that is synchronous across the network (*Kam et al., 2013a*; *Telgkamp and Ramirez, 1999*). Integrated rhythmic preBötC population activity consists of: i) large bursts, reflecting high-frequency action potential firing across a number of neurons, that are coincident with inspiratory motor output as well as ii) smaller deflections, that is burstlets, which reflect lower frequency action potential firing across a similar number of preBötC neurons and which occur without simultaneous motor output (*Figure 1A,G*; *Kam et al., 2013a*). We investigated how opioids affect preBötC burstlets and bursts in neonatal mouse brainstem slices that contain the preBötC and the hypoglossal motor nucleus and nerve (XII) and generate a physiologically relevant motor output, that is inspiratory-related rhythmic activity. To elicit rhythmic inspiratory activity in medullary slices, extracellular K$^+$ is typically raised to 8–9 mM to enhance circuit excitability and compensate for excitatory drive that is lost due to isolation of the preBötC from the intact brainstem (*Del Negro et al., 2009*; *Smith et al., 1991*). Under baseline conditions in artificial cerebrospinal fluid (ACSF) containing 9 mM K$^+$ and 1.5 mM Ca$^{2+}$ ('9/1.5'), preBötC population activity and XII activity, when integrated, consists primarily of coincident, large amplitude rhythmic bursts (*Figure 1A*; *Smith et al., 1991*). Subsaturating concentrations of [D-Ala$^2$,N-MePhe$^4$,Gly-ol$^5$]-enkephalin (DAMGO), a potent synthetic µ-opioid receptor (µOR) agonist, caused a dose-dependent decrease in the frequency (*f*) of rhythmic preBötC and XII bursts (*Figure 1A–C*). 10 nM DAMGO significantly decreased preBötC *f* from 0.22 ± 0.05 Hz to 0.13 ± 0.04 Hz (p=0.01, n = 6; *Figure 1C*). 30 nM DAMGO further decreased preBötC *f* to 0.07 ± 0.04 Hz (p=0.0001, n = 6; *Figure 1C*), and 100 nM DAMGO led to complete cessation of XII activity and, in some cases, also blocked all preBötC activity (*Figure 1B*).

The changes in XII *f* were not caused by a reversion of preBötC bursts to preBötC burstlets (*Kam et al., 2013a*). The fraction of total preBötC events that were burstlets (burstlet fraction), which was very low in control conditions without DAMGO (0.14 ± 0.14), was not significantly different from that in 30 or 100 nM DAMGO (p=0.8, n = 6; *Figure 1A,C*). Burstlet fraction was unaffected by changes in preBötC *f*, as there were no significant correlations in control 9/1.5 ACSF conditions, and no correlations developed in 10 nM or 30 nM DAMGO (p=0.8, n = 6; *Supplementary file 1*). Moreover, the amplitude of preBötC bursts was similar between control conditions without DAMGO

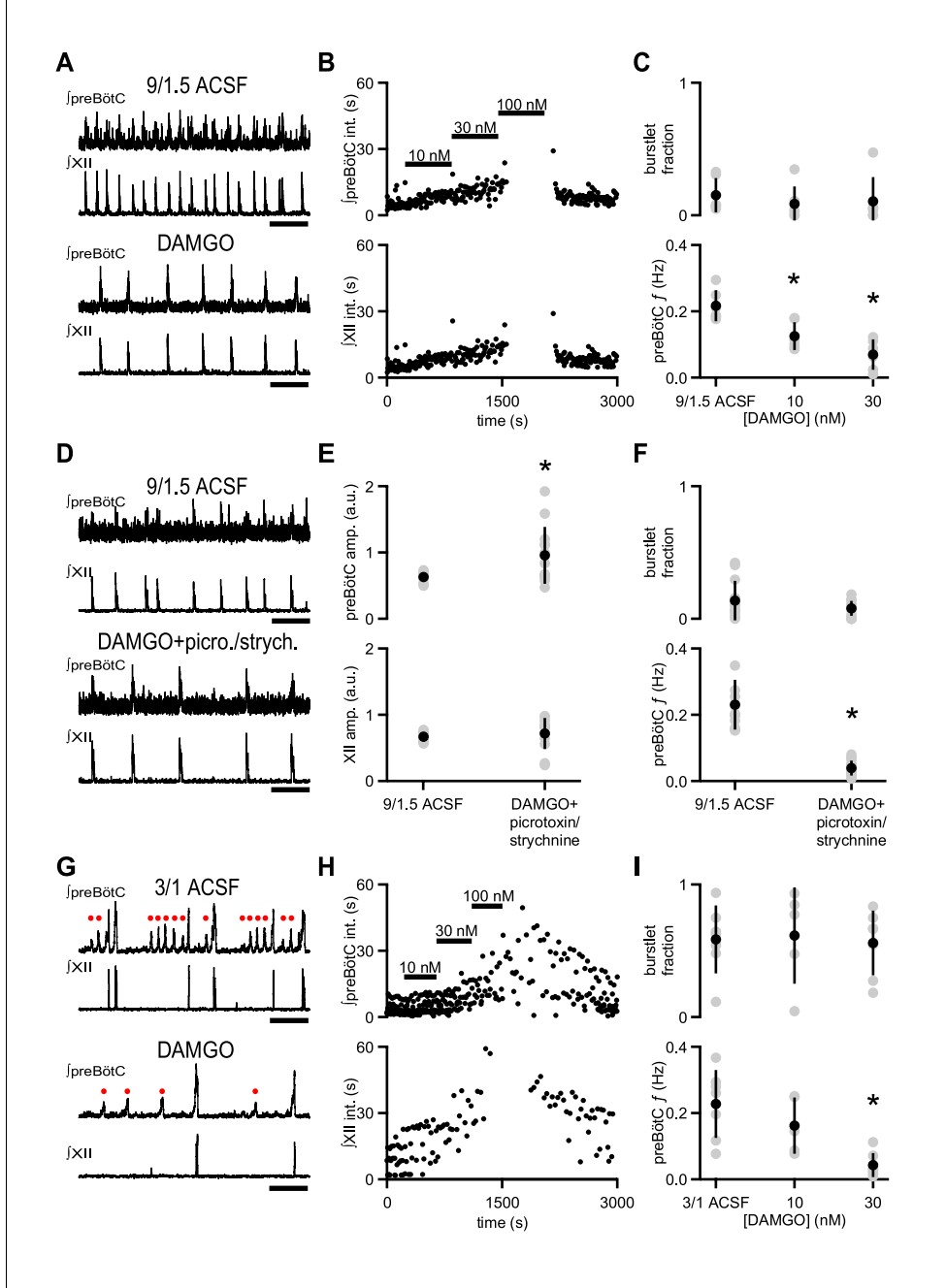

**Figure 1.** μOR activation depresses inspiratory frequency without affecting burstlet fraction. (**A**) Representative traces of ∫preBötC and ∫XII population activity in 9/1.5 ACSF control and 30 nM DAMGO. Scale bar, 10 s. (**B**) Representative time course of an experiment where increasing concentrations of DAMGO were bath-applied in 9/1.5 ACSF. Interevent intervals (int.) in preBötC (top) and XII (bottom) are plotted. In this experiment, 100 nM DAMGO led to cessation of rhythm. (**C**) Average burstlet fraction and preBötC *f* in 9/1.5 ACSF and 10 nM and 30 nM DAMGO. Increasing concentrations of DAMGO did not significantly affect burstlet fraction whereas preBötC *f* was significantly lower in 10 nM and 30 nM DAMGO. *, p<0.05, One-way ANOVA, post-hoc Tukey test, n = 6. (**D**) Representative traces of ∫preBötC and ∫XII population activity in 9/1.5 ACSF control and in 30 nM DAMGO with 100 μM picrotoxin and 1 μM strychnine (picrotoxin/strychnine). Scale bar, 10 s. (**E**) Average preBötC burst and XII burst amplitude (amp.) in 9/1.5 ACSF and in 30 nM DAMGO with 100 μM picrotoxin and 1 μM strychnine (picrotoxin/strychnine). Addition of DAMGO and picrotoxin/strychnine increased preBötC burst amplitude compared to 9/1.5 ACSF control while XII burst amplitudes were not significantly different. The low frequency or lack of preBötC burstlets precluded analysis of their amplitudes. *, p<0.05, Student's *t*-test, n = 13. (**F**) Average

*Figure 1 continued on next page*

*Figure 1 continued*

burstlet fraction and preBötC *f* in 9/1.5 ACSF and in 30 nM DAMGO with 100 μM picrotoxin and 1 μM strychnine (picrotoxin/strychnine). When picrotoxin/strychnine was added with DAMGO, preBötC *f* remained significantly reduced and burstlet fraction was similar. *, p<0.05, Student's *t*-test, n = 13. (**G**) Representative traces of ∫preBötC and ∫XII population activity in 3/1 ACSF control and 30 nM DAMGO. Both small amplitude burstlets (•), which did not produce XII activity, and large amplitude bursts, which generated XII bursts, were observed. Scale bar, 10 s. (**H**) Representative time course of an experiment, where increasing concentrations of DAMGO were bath-applied in 3/1 ACSF. Interevent intervals in preBötC (top) and XII (bottom) are plotted. In this experiment, 100 nM DAMGO led to cessation of XII activity while preBötC rhythm persisted. (**I**) Average burstlet fraction and *f* in 3/1 ACSF and 10 nM and 30 nM DAMGO. Increasing concentrations of DAMGO did not significantly affect burstlet fraction whereas preBötC *f* was significantly lower in 30 nM DAMGO. *, p<0.05, One-way ANOVA, post-hoc Tukey test, n = 7.

(0.60 ± 0.10 a.u.) and 9/1.5 ACSF with 10 nM (0.61 ± 0.13 a.u.) or 30 nM DAMGO (0.56 ± 0.19 a.u.; p=0.8, n = 6).

In previous studies in rat medullary slices bathed in 9/1.5 ACSF, DAMGO-induced decreases in XII *f* were not affected by the presence of blockers of inhibitory synaptic transmission (*Gray et al., 1999*), which can raise excitability independent of baseline increases in extracellular $K^+$ (*Del Negro et al., 2009*; *Ren and Greer, 2006*; *Shao and Feldman, 1997*). We therefore examined whether the effects of DAMGO were similarly persistent when a cocktail of the $GABA_A$ receptor antagonist picrotoxin and the glycine receptor antagonist strychnine (picrotoxin/strychnine) was added to mouse medullary slices and whether recording preBötC population activity could reveal effects that were not visible in XII activity (*Del Negro et al., 2009*; *Kam et al., 2013a*). In 9/1.5 ACSF, adding picrotoxin/strychnine (100 μM/1 μM) with 30 nM DAMGO increased preBötC burst amplitude (0.95 ± 0.43 a.u.) compared to baseline (0.63 ± 0.062 a.u.; p=0.01, n = 13; *Figure 1D,E*). Despite the increase in preBötC burst amplitude, preBötC *f* in slices bathed in 30 nM DAMGO and picrotoxin/strychnine (0.038 ± 0.023 Hz) was similar to preBötC *f* in 30 nM DAMGO alone (0.073 ± 0.037 Hz; p=0.6, n = 13) and significantly lower than preBötC *f* in 9/1.5 ACSF (0.23 ± 0.075 Hz; p=4×10$^{-6}$, n = 13; *Figure 1D,F*). preBötC *f* and burst amplitude showed no correlation under these conditions (*Supplementary file 1*). Burstlet fraction was again unaltered when DAMGO and picrotoxin/strychnine were co-applied (p=0.09, n = 13; *Figure 1D,F*), and no significant correlation between burstlet fraction and preBötC *f* was observed (*Supplementary file 1*).

To determine the effects of DAMGO when preBötC burstlets could be separated from bursts, we lowered ACSF $K^+$ and $Ca^{2+}$ concentrations to physiological levels (*Kam et al., 2013a*). In 3 mM $K^+$ and 1 mM $Ca^{2+}$ ('3/1') ACSF, preBötC rhythmic activity is maintained, but consists of a mixed pattern of burstlets and bursts (*Figure 1G*, upper traces). While the burstlet fraction under these conditions (0.58 ± 0.26) was significantly higher than that in 9/1.5 ACSF (0.14 ± 0.14; p=0.003, n = 6 for 9/1.5, n = 7 for 3/1), preBötC *f* was similar (0.23 ± 0.10 Hz in 3/1 vs. 0.22 ± 0.05 Hz in 9/1.5; p=0.8, n = 6 for 9/1.5, n = 7 for 3/1; *Figure 1C,I*). XII *f* in 3/1 ACSF (0.08 ± 0.05 Hz), on the other hand, was significantly lower than XII *f* in 9/1.5 ACSF (0.2 ± 0.06 Hz; p=0.002, n = 6 for 9/1.5, n = 7 for 3/1; *Figure 1B,H*), a consequence of the increased fraction of preBötC burstlets that were not transmitted to XII output.

In 3/1 ACSF, DAMGO decreased preBötC *f* in a dose-dependent manner, similar to its effects in 9/1.5 ACSF. Addition of 30 nM DAMGO significantly reduced preBötC *f* from 0.22 ± 0.1 Hz to 0.04 ± 0.04 Hz (p=0.001, n = 7; *Figure 1G–I*). This reduction was not significantly different from the effects of 30 nM DAMGO in 9/1.5 ACSF (p=0.2, n = 6 for 9/1.5, n = 7 for 3/1; *Figure 1C,I*). Higher concentrations of DAMGO (≥100 nM) frequently blocked all preBötC population activity. In 3/1 ACSF, burstlet and burst amplitudes (burstlet: 0.38 ± 0.05 a.u.; burst: 0.73 ± 0.08 a.u.) were similar to those after addition of 10 nM DAMGO (burstlet: 0.36 ± 0.04 a.u.; burst: 0.78 ± 0.10 a.u.) and 30 nM DAMGO (burstlet: 0.34 ± 0.11 a.u.; burst: 0.65 ± 0.21 a.u.; burstlet: p=0.7, burst: p=0.3, n = 7). Importantly, the burstlet fraction was again unaffected by DAMGO (p=1.0, n = 7; *Figure 1I*). Burstlet fraction and preBötC *f* showed no significant correlations across conditions in 3/1 ACSF (p=0.6, n = 7; *Supplementary file 1*).

To determine whether opioids specifically affected rhythmogenic activity independent of burst production, we used low concentrations of $Cd^{2+}$ (6–25 μM; *Kam et al., 2013a*) to increase the burstlet

fraction significantly from 0.42 ± 0.30 to 0.78 ± 0.20 (p=0.02, n = 8) without changing preBötC $f$ or burstlet amplitude ($f$: p=0.7, amplitude: p=0.06, n = 8; *Figure 2A,B*). $Cd^{2+}$-induced decreases in excitability and synaptic efficacy (*Kam et al., 2013a*; *Lu et al., 2007*) resulted in an increased sensitivity to DAMGO. Bath application of 1 nM DAMGO in these conditions was sufficient to abolish preBötC rhythmic activity completely in 7 of 8 slices. Significantly lower concentrations of DAMGO (0.01–10 nM) slowed preBötC $f$ from 0.28 ± 0.09 Hz to 0.18 ± 0.07 Hz (p=0.001, n = 8; *Figure 2A,B*). This relative decrease in preBötC $f$ (0.66 ± 0.16) was similar to the effect of 10 nM DAMGO on preBötC $f$ in 3/1 ACSF with no $Cd^{2+}$ (0.63 ± 0.16; p=0.8, n = 5 for 3/1, n = 8 for $Cd^{2+}$). A statistically significant

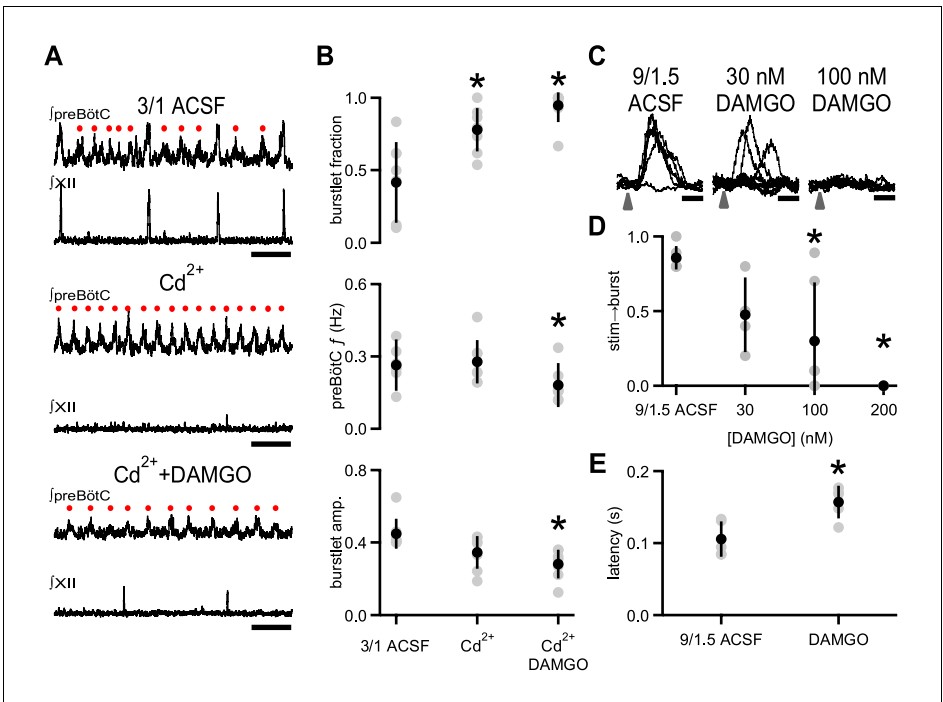

**Figure 2.** DAMGO specifically modulates preBötC rhythmogenesis. (**A**) Representative traces of ∫preBötC and ∫XII population activity in 3/1 ACSF control, 8 μM $Cd^{2+}$, and 8 μM $Cd^{2+}$ + 10 nM DAMGO. In 3/1 ACSF (top), rhythmic activity consisted of both burstlets (•) and bursts. After $Cd^{2+}$ application (middle), preBötC bursts were no longer observed, and XII activity was abolished, leaving only a preBötC burstlet rhythm with a $f$ similar to control. With addition of DAMGO (bottom), burstlet $f$ and amplitude decreased. Note: the two large deflections in ∫XII (bottom) are not synchronized with preBötC activity, and are likely non-inspiratory or electrical artifact. Scale bar, 5 s. (**B**) Average burstlet fraction, preBötC $f$, and burstlet amplitude (amp.) in 3/1 ACSF control, 6–25 μM $Cd^{2+}$, and 6–25 μM $Cd^{2+}$ + 0.1–10 nM DAMGO. Addition of $Cd^{2+}$ significantly increased the burstlet fraction while preBötC $f$ and burstlet amplitudes were unchanged. Bath application of DAMGO significantly decreased preBötC $f$ (consisting almost solely of burstlets) and burstlet amplitude. *, p<0.05, one-way ANOVA, post-hoc Tukey test, n = 8. (**C**) Overlays of representative traces of ∫preBötC from successive photoexcitation of 8 Dbx1[+] neurons in control 9/1.5 ACSF, 30 nM DAMGO, and 100 nM DAMGO. The illumination pattern consisted of 10 μm soma-centered spots over the targeted neurons. Photoexcitation was provided by 405 nm holographic illumination of 500 μM MNI-glutamate. Arrow represents time of laser stimulation. In 9/1.5 ACSF, a failure was seen, but burst initiation was otherwise reliably successful. Bursts were consistently triggered after a ~ 100 ms latency. In 30 nM DAMGO, failures were more frequent when the same stimulation parameters were used, and the latency between laser stimulation and burst initiation was longer and more variable. In 100 nM DAMGO, bursts in this experiment were no longer triggered with the same stimulation parameters. Scale bar, 200 ms. (**D**) Average success rate (stim→burst) for burst initiation in control 9/1.5 ACSF conditions and in increasing concentrations of DAMGO using entraining stimuli that elicits >80% success in 9/1.5 ACSF. To entrain the rhythm, 5–10 Dbx1[+] neurons were stimulated 8–20 times every 3–6 s. DAMGO produced a dose-dependent decrease in the success rate for triggering bursts with a significant reduction in success between control and at 100 nM and 200 nM DAMGO. *, p<0.05, one-way ANOVA, post-hoc Tukey test, n = 6. (**E**) Average latency between laser stimulation and burst initiation when bursts were successfully triggered during entraining stimuli was increased in 30 nM DAMGO compared to control 9/1.5 ACSF. *, p<0.05, Student's $t$-test, n = 5.

correlation between burstlet fraction and preBötC $f$ was observed in the baseline 3/1 ACSF condition, but was not present in the $Cd^{2+}$ alone or $Cd^{2+}$ and DAMGO conditions, and the differences in linear relationships across the conditions were not statistically significant (p=0.9, n = 8; *Supplementary file 1*). The presence of both $Cd^{2+}$ and DAMGO did, however, reduce burstlet amplitudes compared to 3/1 ACSF alone (3/1 ACSF: 0.45 ± 0.08 a.u., $Cd^{2+}$+DAMGO: 0.28 ± 0.08 a.u.; p=0.005, n = 8; *Figure 2A, B*).

Depression of preBötC burstlet $f$ by µOR activation across several conditions points to DAMGO modulating an emergent preBötC rhythmogenic mechanism. We previously hypothesized that triggering synchronous activity in a small number of neurons could initiate or accelerate the same assembly mechanism underlying burstlets and endogenous rhythm generation. To elicit simultaneous activity in a defined number of preBötC neurons, we used holographic photolysis of a caged glutamate compound (MNI-glutamate), which produces bursts of action potentials that resemble endogenous inspiratory bursts in the targeted neurons (*Kam et al., 2013b*). With this technique, targeted photoexcitation of 4–9 preBötC inspiratory neurons of unspecified molecular identity could trigger ectopic bursts with a latency that we hypothesized represents network assembly (*Feldman and Kam, 2015*; *Kam et al., 2013b*). Here, we targeted preBötC Dbx1-derived (Dbx1$^+$) neurons, a glutamatergic subpopulation essential for respiratory rhythm generation (*Bouvier et al., 2010*; *Gray et al., 2010*; *Wang et al., 2014*), for holographic photostimulation to determine if opioids modulate this rhythmogenesis-related process (*Figure 2C–E*). In slices from Dbx1$^+$ reporter (Dbx1$^{cre}$;Rosa26$^{tdTomato}$) mice (*Kottick et al., 2017*) bathed in 9/1.5 ACSF and 500 µM MNI-glutamate, ectopic bursts were reliably elicited when 5–10 Dbx1$^+$ neurons were targeted (cumulative probability of success: 0.86 ± 0.08, n = 6; *Figure 2C,D*). When a set of neurons that reliably resulted in bursts in response to photostimulation in control conditions were then excited in 30 nM DAMGO, the probability of eliciting a burst was reduced in some slices (cumulative probability of success: 0.48 ± 0.25, p=0.12, n = 4; *Figure 2C,D*). In 100 nM DAMGO, the success rate decreased significantly as stimulation of the threshold set that had previously reliably triggered bursts was further compromised and, in two out of six slices, failed to trigger a single burst (cumulative probability of success: 0.30 ± 0.39, p=0.006, n = 6; *Figure 2C,D*). In 200 nM DAMGO, stimulation could not elicit bursts in any slice (p=0.0003, n = 6; *Figure 2C,D*). Importantly, the latency between laser stimulation and burst initiation in cases when bursts were triggered in DAMGO (157 ± 22 ms) was significantly increased compared to that in baseline conditions (105 ± 24 ms; p=0.002, n = 5; *Figure 2E*). These data are consistent with our hypothesis that burstlets and the latency between photostimulation and burst initiation reflect the same underlying emergent rhythmogenic process and are both modulated by µOR activation.

## µOR-expressing Dbx1$^+$ neurons mediate the rhythmogenic effects of opioids

We hypothesized that DAMGO modulated these rhythmogenic mechanisms by acting directly on preBötC Dbx1$^+$ neurons that contain µOR mRNA (*Hayes et al., 2017*). We first sought to determine whether these neurons express µOR protein. In Dbx1$^{Cre}$;Rosa26$^{tdTomato}$ mice, µOR protein expression overlapped with the spatial distribution of tdTomato-expressing Dbx1$^+$ neurons in preBötC and the adjacent intermediate band of the reticular formation (IRt; *Figure 3A*) and was localized to individual preBötC Dbx1$^+$ neurons (*Figure 3B,C*).

To test whether µORs in Dbx1$^+$ neurons mediated DAMGO inhibition, we used a Dbx1$^{Cre}$;Oprm1$^{fl/fl}$ mouse line (*Weibel et al., 2013*), in which µORs are genetically deleted in Dbx1$^+$ neurons. We first examined whether preBötC µOR expression was knocked down in P2 Dbx1$^{Cre}$;Oprm1$^{fl/fl}$ mice compared to their littermate controls (*Figure 3D–I*). Direct visualization of Dbx1$^+$ neurons in these mice is difficult as Dbx1 is not expressed postnatally (*Bouvier et al., 2010*; *Gray et al., 2010*). Another preBötC marker is the neurokinin-1 receptor (NK1R), but NK1R protein is predominantly found in neuronal processes (*Tan et al., 2010*) while NK1R mRNA is sparsely expressed in somata (*Figure 3—figure supplement 1*). We therefore used tachykinin precursor 1 peptide (TAC1) expression, which colocalizes with NK1R, ventral to nucleus ambiguus (NA) as a marker for preBötC (*Figure 3—figure supplement 1*; *Hayes et al., 2017*). To measure µOR and TAC1 expression, we performed fluorescence in situ hybridization, which avoids the non-specific labeling that can occur with some µOR antibodies (*Schmidt et al., 2013*) and allows localization of TAC1 expression to somata (*Figure 3—figure supplement 1*). Abundant (>15 puncta) TAC1 mRNA was found in larger

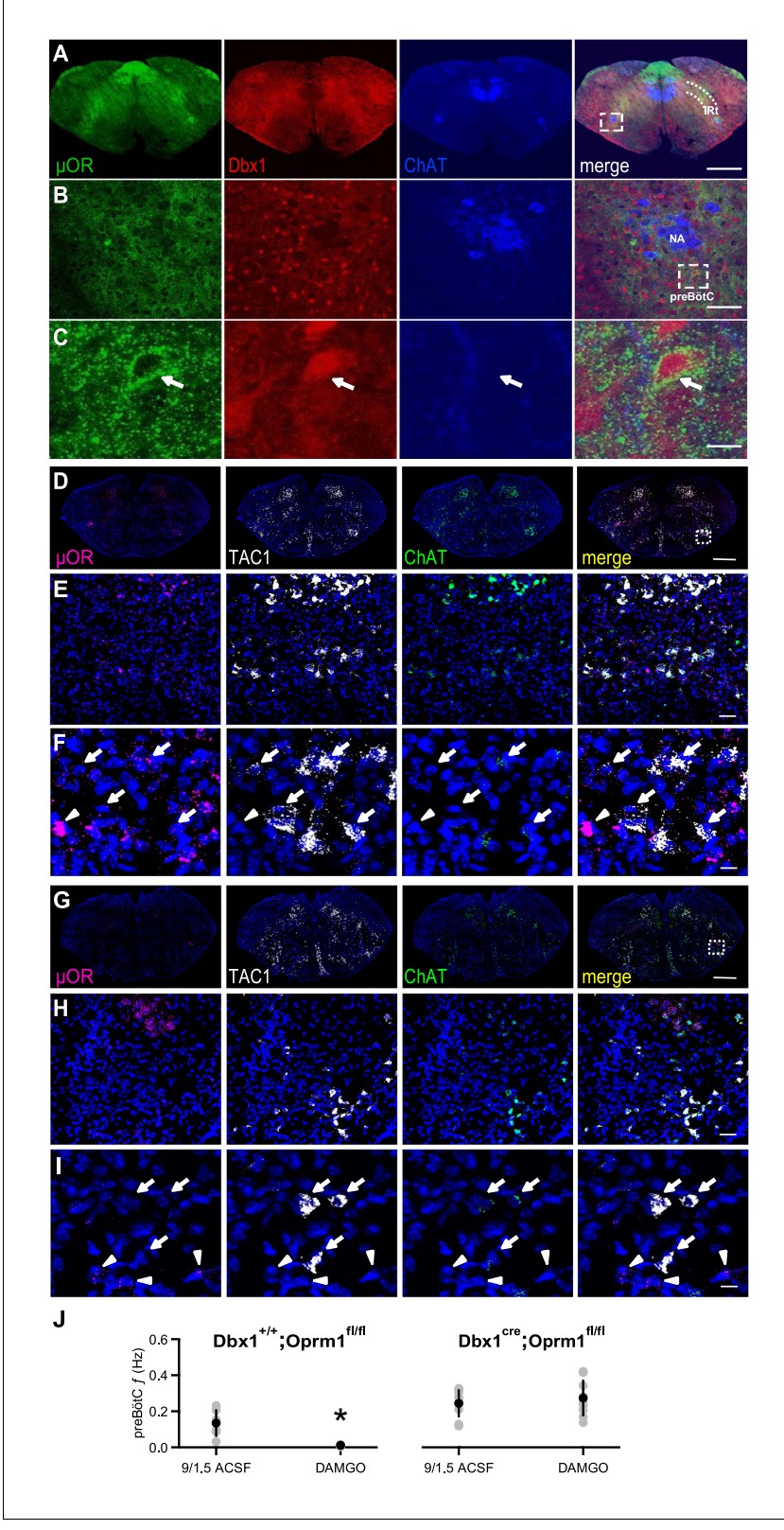

**Figure 3.** µORs expressed in Dbx1[+] neurons mediate DAMGO inhibition of preBötC rhythmic activity. (**A**) Confocal images of brainstem sections of a *Dbx1^{Cre};Rosa26^{tdTomato}* mouse pup (P2) double-stained with antibodies to Choline acetyltransferase (ChAT; blue) and µOR (green). Coronal sections showing µOR expression in the preBötC and the Intermediate band of the Reticular Formation (IRt) and colocalization with Dbx1[+] neurons

*Figure 3 continued on next page*

*Figure 3 continued*

in the preBötC. Scale bar, 500 µm. (B) Higher magnification of nucleus ambiguus (NA) and preBötC from the boxed region in A. Scale bar, 50 µm. (C) Expanded images of the boxed preBötC region highlighted in B. The arrow shows a preBötC Dbx1$^+$ neuron with abundant punctate µOR expression on the cell body. Scale bar, 10 µm. (D) Confocal image of a medullary brainstem section at the level of the preBötC from a control P2 *Dbx1$^{+/+}$; Oprm1$^{fl/fl}$* mouse processed with RNAScope. Probes for µOR (magenta), tachykinin precursor 1 peptide (TAC1; white), and ChAT (green) were used, and the tissue was counterstained with DAPI (blue). A box marks the location of preBötC. Scale bar, 1 mm. (E) Higher magnification of the boxed preBötC region in D. µOR and TAC1 were highly expressed in this region. Scale bar, 100 µm. (F) Higher magnification images of TAC1-positive neurons and µOR mRNA (arrows). Colocalization of TAC1 and multiple µOR puncta was observed in several neurons (arrows) as well as abundant µOR puncta in negative TAC1 regions (arrowhead). Scale bar, 40 µm. (G) Confocal image of a medullary brainstem section at the level of the preBötC from a P2 *Dbx1$^{cre/cre}$;Oprm1$^{fl/fl}$* mouse processed with RNAScope. Probes for µOR (magenta), TAC1 (white), and ChAT (green) were used, and the tissue was counterstained with DAPI (blue). In the preBötC of these mice, the density of µOR puncta was low. A box marks the location of preBötC. Scale bar, 1 mm. (H) Higher magnification of the boxed preBötC region in G. The density of µOR puncta was low, and µOR expression was not associated with TAC1 expression. Scale bar, 100 µm. (I) Higher magnification images of TAC1-expressing neurons from H with few to no µOR puncta, as well as µOR transcripts without TAC1 expression (arrowheads). Scale bar, 40 µm. (J) Average *f* of preBötC activity in control 9/1.5 conditions and 100 nM DAMGO in *Dbx1$^{Cre}$;Oprm1$^{fl/fl}$* mice (right), where µORs are selectively deleted in Dbx1$^+$ neurons, and their *Dbx1$^{+/+}$;Oprm1$^{fl/fl}$* littermate controls with preserved µOR expression (left). Whereas 100 nM DAMGO reduced preBötC *f* significantly in *Dbx1$^{+/+}$;Oprm1$^{fl/fl}$* control mice, the same concentration had no effect on *f* when µORs were deleted. *, p<0.05, Student's *t*-test, n = 6 for *Dbx1$^{+/+}$;Oprm1$^{fl/fl}$*, n = 9 for *Dbx1$^{Cre}$; Oprm1$^{fl/fl}$*.

The online version of this article includes the following figure supplement(s) for figure 3:

**Figure supplement 1.** µOR, TAC1, and NK1R expression in preBötC.

neurons within the preBötC (soma diameter: 27.1 ± 4.3 µm, n = 284 neurons from two mice). Choline acetyltransferase (ChAT) mRNA probes were used to mark NA. In *Dbx1$^{+/+}$;Oprm1$^{fl/fl}$* control mice, approximately 90% of TAC1-positive neurons in the preBötC co-expressed µOR and NK1R mRNA (*Figure 3—figure supplement 1*), and TAC1 and µOR mRNA expression (*Figure 3D–F*) resembled the immunohistochemical distribution of Dbx1$^+$ neurons in the Dbx1$^+$ reporter mice (*Figure 3A–C*). Compared with µOR expression in *Dbx1$^{+/+}$;Oprm1$^{fl/fl}$* mice (*Figure 3D–F*), µOR expression in *Dbx1$^{Cre}$;Oprm1$^{fl/fl}$* mice was significantly reduced in TAC1-positive preBötC neurons (p=0.0001, *Dbx1$^{+/+}$;Oprm1$^{fl/fl}$*: H-score = 139.07 ± 39.33, n = 5 mice; *Dbx1$^{Cre}$;Oprm1$^{fl/fl}$*: H-score = 47.98 ± 19.77, n = 3 mice; *Figure 3G–I*).

Having confirmed a decrease in µOR expression in the preBötC of *Dbx1$^{Cre}$;Oprm1$^{fl/fl}$* mice, we determined whether the functional effects of DAMGO on preBötC rhythm were consequently reduced. In rhythmic slices from *Dbx1$^{Cre}$;Oprm1$^{fl/fl}$* mice in 9/1.5 ACSF, 100 nM DAMGO had little effect on preBötC *f* (9/1.5: 0.24 ± 0.08 Hz, DAMGO:=0.27 ± 0.1 Hz, p=0.2, n = 9; *Figure 3J*). In contrast, preBötC *f* was significantly reduced by 100 nM DAMGO in slices from *Dbx1$^{+/+}$;Oprm1$^{fl/fl}$* cre-negative littermates (9/1.5: 0.13 ± 0.08 Hz, DAMGO:=0.011 ± 0.005 Hz, p=0.01, n = 6; *Figure 3J*). These data indicate that DAMGO specifically acts on µORs expressed on preBötC Dbx1$^+$ neurons to modulate rhythmogenesis.

## DAMGO-mediated depression is diminished by Substance P

The coexpression of µOR and NK1R in preBötC Dbx1$^+$ neurons raises the possibility that NK1R activation, which increases respiratory frequency in wild type rodents (*Gray et al., 1999*; *Yeh et al., 2017*), could be effective in reducing the depressant effects of DAMGO. In 3/1 ACSF, 30 nM DAMGO was much less effective in reducing rhythmic activity in the presence of the NK1R agonist Substance P (SP; 500 nM). preBötC *f* when SP and DAMGO were both present (0.17 ± 0.05 Hz) was not significantly different from that in control 3/1 ACSF conditions, that is in the absence of SP and DAMGO (0.24 ± 0.12 Hz; p=0.09, n = 6; *Figure 4A,B*). Indeed, the normalized change in preBötC *f* when SP was applied with 30 nM DAMGO in 3/1 ACSF (0.82 ± 0.15) was greater than that in 9/1.5 ACSF (0.32 ± 0.22), 9/1.5 ACSF with picrotoxin/strychnine (0.19 ± 0.14), and 3/1 ACSF in the presence of DAMGO (0.19 ± 0.11; p=$2\times10^{-6}$, n = 6 for 9/1.5, n = 7 for 3/1, n = 13 for picrotoxin/strychnine, n = 6 for SP; *Figure 4B*). Normalized burstlet fraction, on the other hand, was similar across conditions (p=0.07, n = 6 for 9/1.5,

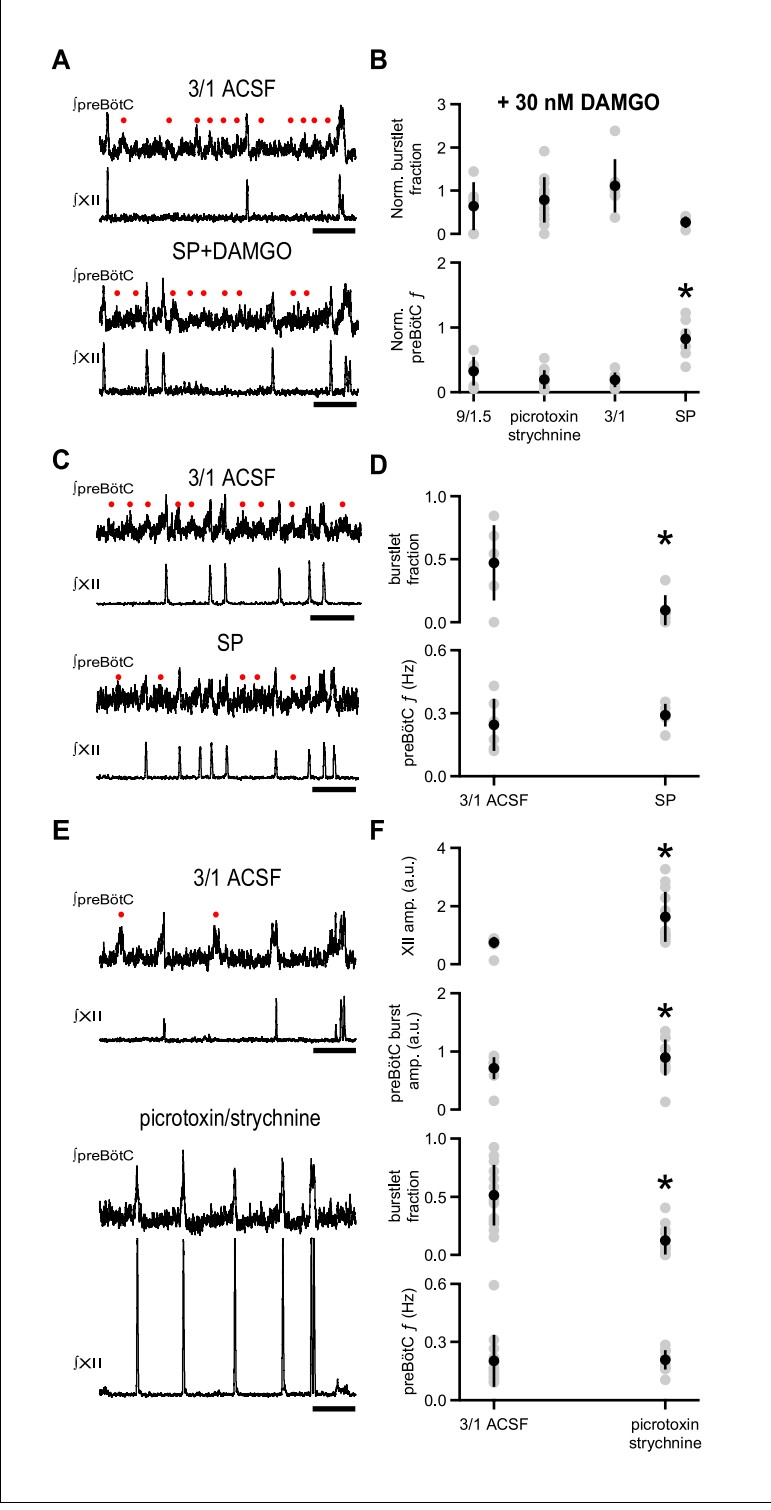

**Figure 4.** Substance P, but not blockade of inhibitory synaptic transmission, relieves DAMGO-mediated depression.  (**A**) Representative traces of ∫preBötC and ∫XII population activity in 3/1 ACSF control and 500 nM SP and 30 nM DAMGO. SP mitigated the effects of DAMGO, and preBötC activity, consisting of both burstlets (•) and bursts, resembled that in control conditions. Scale bar, 10 s. (**B**) Average normalized burstlet fraction and preBötC *f* when 30 nM DAMGO was added alone (9/1.5, 3/1) or following application of either 100 µM picrotoxin/1 µM strychnine or 500 nM SP. For each experiment, values were normalized to burstlet fraction or preBötC *f* during baseline 9/1.5 or 3/1 ACSF in that experiment. The normalized preBötC *f* when SP was added with DAMGO was
*Figure 4 continued on next page*

*Figure 4 continued*

significantly higher than all other conditions. *, p<0.05, one-way ANOVA, post-hoc Tukey test, n = 6 for 9/1.5, n = 7 for 3/1, n = 13 for picrotoxin/strychnine, n = 6 for SP. (C) Representative traces of ∫preBötC and ∫XII population activity in 3/1 ACSF control and 500 nM SP. In 3/1 ACSF, rhythmic activity consisted of both burstlets (•) and bursts. After SP, *f* was unchanged, but more preBötC bursts were observed. Scale bar, 10 s. (D) Average burstlet fraction and *f* in 3/1 ACSF control condition and 500 nM SP. SP significantly decreased the burstlet fraction, but did not change preBötC *f*. *, p<0.05, Student's *t*-test, n = 13. (E) Representative traces of ∫preBötC and ∫XII population activity in 3/1 ACSF control and 100 μM picrotoxin/1 μM strychnine. In 3/1 ACSF, rhythmic activity consisted of both burstlets (•) and bursts. After picrotoxin/strychnine, *f* was unchanged; however, more preBötC bursts were observed, and XII bursts were larger. Scale bar, 10 s. (F) Average burstlet fraction and *f* in 3/1 ACSF control condition and 100 μM picrotoxin/1 μM strychnine. Picrotoxin/strychnine significantly decreased the burstlet fraction, but did not change preBötC *f*. *, p<0.05, Student's *t*-test, n = 13.

n = 7 for 3/1, n = 13 for picrotoxin/strychnine, n = 6 for SP; *Figure 4B*). While a statistically significant decrease in preBötC burst amplitude was observed in SP and DAMGO (0.50 ± 0.16 a.u.) compared to 3/1 ACSF (0.67 ± 0.17 a.u.; p=0.005, n = 6), preBötC burstlet and XII burst amplitudes in SP and DAMGO (burstlet: 0.48 ± 0.14 a.u.; XII: 0.79 ± 0.25 a.u.) were similar to those in control conditions (burstlet: 0.56 ± 0.10 a.u.; XII: 0.80 ± 0.07 a.u.; burstlet: p=0.2, XII: p=0.9, n = 6).

We next examined whether SP alone affected preBötC *f* in 3/1 ACSF. Bath-applied SP (500 nM) significantly reduced the burstlet fraction from 0.47 ± 0.3 to 0.10 ± 0.12 (p=0.01, n = 6), but did not affect preBötC *f* (3/1 ACSF: 0.24 ± 0.12 Hz, SP: 0.29 ± 0.05, p=0.4, n = 6; *Figure 4C,D*). No significant correlations between burstlet fraction and preBötC *f* were observed when SP was applied (p=0.3, n = 6; *Supplementary file 1*). preBötC burst and burstlet and XII burst amplitudes did not change significantly in SP compared to 3/1 ACSF (preBötC burstlet: p=0.3, preBötC burst: p=0.07, XII: p=0.8, n = 6).

We compared the effects of SP to those of bath-applied picrotoxin/strychnine (100 μM/1 μM) in 3/1 ACSF. Picrotoxin/strychnine in 3/1 ACSF also decreased burstlet fraction (3/1 ACSF: 0.51 ± 0.26, picrotoxin/strychnine: 0.12 ± 0.12, p=0.0003, n = 13) and did not significantly alter preBötC *f* (3/1 ACSF: 0.20 ± 0.13 Hz, picrotoxin/strychnine: 0.21 ± 0.05 Hz, p=0.9, n = 13; *Figure 4E,F*). Again, no significant correlations were observed between burstlet fraction and preBötC *f* in picrotoxin/strychnine, and the slope of the linear fit was similar to that in control 3/1 ACSF (p=0.4, n = 13; *Supplementary file 1*). While preBötC burstlet amplitudes were similar across these conditions (3/1 ACSF: 0.61 ± 0.13 a.u., picrotoxin/strychnine: 0.74 ± 0.42 a.u., p=0.2, n = 13), preBötC and XII burst amplitudes were significantly larger in picrotoxin/strychnine (preBötC burst: 0.90 ± 0.31 a.u., XII burst: 1.63 ± 0.86 a.u.) compared to 3/1 ACSF (preBötC burst: 0.71 ± 0.19 a.u., XII burst: 0.74 ± 0.20 a.u.; preBötC burst: p=0.004, XII burst: p=0.003, n = 13; *Figure 4E,F*). No significant correlations were observed between preBötC burst amplitude and preBötC *f* in picrotoxin/strychnine, and the slope of this linear fit was similar to 3/1 ACSF (p=0.2, n = 13; *Supplementary file 1*). Therefore, while both SP and inhibitory blockade can modulate the conversion of burstlets to bursts and picrotoxin/strychnine can have an effect on preBötC burst amplitudes, only NK1R activation could offset the depressant effects of DAMGO.

## Discussion

Inspiratory rhythm generation in mammals must be both robust, to operate almost continuously throughout the lifetime, and highly labile, to adapt to changing metabolic and environmental conditions and coordinate with other behaviors. How the preBötC generates rhythm to meet these challenges has thus far defied a thorough understanding. We put forth a hypothesis that respiratory rhythmogenesis is driven not by strong bursts, but by an emergent process that manifests as preBötC burstlets (*Kam et al., 2013a*). In contrast to pacemaker-driven networks (*Phillips et al., 2019*), where rhythm depends on the kinetics of conductances, such as $I_h$ or persistent inward currents (*Marder and Calabrese, 1996*), in this model, the period is determined by the gradual synchronization of spontaneously active, recurrently connected excitatory preBötC neurons (*Ashhad and Feldman, 2019*; *Feldman and Kam, 2015*). Each cycle consists of the generation, propagation, and

integration of spontaneous activity within the preBötC and culminates in the generation of a burstlet (*Ashhad and Feldman, 2019*; *Feldman and Kam, 2015*).

A strong prediction of this hypothesis is that manipulations that depress respiratory frequency should also slow burstlet *f*. Indeed, we found that DAMGO specifically slowed preBötC *f* without affecting burstlet fraction when the rhythm was: i) composed of mostly bursts; ii) a mixed pattern of bursts and burstlets; or iii) mostly (or all) burstlets. Additionally, DAMGO prolonged the latency between photostimulation of $\leq$10 Dbx1[+] neurons and burst generation. Together, these effects are consistent with our hypothesis that both are products of percolation of activity within the preBötC microcircuit, a critical process for rhythmogenesis (*Kam et al., 2013a*; *Kam et al., 2013b*). Moreover, when μORs were genetically deleted in Dbx1[+] neurons in *Dbx1^Cre^;Oprm1^fl/fl^* mice, the potent rhythmic slowing effect of DAMGO in slices was no longer present, suggesting that DAMGO directly affects rhythmogenic mechanisms through μORs in preBötC Dbx1[+] neurons. In line with our functional studies, we found abundant μOR expression in preBötC Dbx1[+] neurons that form the core circuitry for inspiratory rhythmogenesis (*Wang et al., 2014*).

How could μOR activation affect the percolation of activity in the preBötC to slow down inspiratory rhythm? The synchronous low-frequency firing across preBötC neurons that defines a burstlet requires that a critical number of preBötC neurons be active and that their firing coincide in a short time frame to allow integration of inputs, depolarization above threshold, and further propagation of activity within the preBötC network (*Feldman and Kam, 2015*). μORs are found on dendrites and cell bodies, where they can regulate neuronal excitability, as well as on axon terminals, where they can inhibit neurotransmitter release via activation of $K^+$ conductances and/or inhibition of $Ca^{2+}$ conductances (*Le Merrer et al., 2009*; *Mansour et al., 1988*; *Montandon et al., 2016*; *Williams et al., 2001*). We suggest that DAMGO decreases the spontaneous firing rate and interferes with propagation of action potentials by hyperpolarizing membrane potential and/or decreasing synaptic probability of release in and among μOR-expressing preBötC Dbx1[+] neurons. These effects prolong the time required for preBötC neurons to increase their firing and reduce the probability that a critical number of neurons is simultaneously active to generate the next burstlet (*Ashhad and Feldman, 2019*; *Feldman and Kam, 2015*; *Kam et al., 2013a*). Thus, by depressing excitability and affecting percolation, DAMGO alters rhythmogenesis, independent of effects on strong bursts of activity associated with motor output.

DAMGO-mediated depression of preBötC *f* was mitigated by the NK1R agonist, SP, an effect that we suggest is mediated by intracellular signaling pathway interactions in preBötC Dbx1[+] neurons that coexpress μOR and NK1R. The μOR expression and DAMGO sensitivity of NK1R-expressing neurons are well-documented (*Gray et al., 1999*; *Montandon et al., 2011*), and $G_s$-protein coupled signaling of NK1R can oppose the $G_{i/o}$-coupled signaling of μORs (*Gray et al., 1999*; *Johnson et al., 1996*; *Manzke et al., 2003*). Indeed, we showed colocalization of μOR with NK1R in some preBötC neurons as well as with a partially overlapping preBötC subpopulation expressing TAC1 that may be one endogenous source of SP (*Liu et al., 2004*). Consistent with increases in XII and preBötC *f* observed when SP is applied to slices bathed in 9/1.5 ACSF (*Gray et al., 1999*; *Yeh et al., 2017*), the effects of co-application of opioids and SP support a role for NK1R activation in modulating rhythm. However, the actions of SP and the interaction of NK1R and μOR can be complicated. In 3/1 ACSF alone, SP did not change preBötC *f*. A ceiling effect, that is preBötC *f* being near or at maximal, may have limited the rhythmogenic effect of SP in 3/1 ACSF alone. Alternatively, the primary targets of NK1R modulation of preBötC, whether within preBötC or elsewhere in the slice, for example medullary raphe, may be state/condition-dependent (*Nattie and Li, 2008*). Additionally, opioids may inhibit SP release, and SP potentiates the antinociceptive effects of morphine through release of endogenous opioids (*Fukazawa et al., 2007*; *Kream et al., 1993*). SP may also act directly on μORs as a weak agonist since naloxone, the μOR antagonist, blocks SP excitation in some neurons (*Davies and Dray, 1977*). Nonetheless, we predict that only neuropeptides and neuromodulators whose receptors are coexpressed with μORs in rhythmogenic preBötC neurons, like NK1Rs, are able to modulate μOR signaling intracellularly to mitigate DAMGO-mediated depression.

Consistent with this hypothesis, the depressant effects of DAMGO were not reduced by inhibitory blockade. The effects of picrotoxin and strychnine on excitability depend on developmental age, extracellular $K^+$, and experimental preparation (*Ren and Greer, 2006*), which may account for their inability to mitigate DAMGO-induced depression. Alternatively, as inhibitory synaptic activity in

preBötC is dispensable for generation of breathing rhythm in vivo and in vitro (*Janczewski et al., 2013*; *Shao and Feldman, 1997*), inhibition may act elsewhere in the microcircuit or upstream of opioidergic signaling pathways within Dbx1$^+$ neurons (*Oke et al., 2018*).

In the absence of DAMGO, we observed that both inhibitory blockade and NK1R activation could specifically affect burstlet fraction or burst amplitude independent of preBötC *f*. Indeed, variability in preBötC *f* was poorly correlated with variability in burstlet fraction across a number of conditions. How could burstlet fraction be specifically modulated? To generate a burst, the low level synchronous firing across the population that defines burstlets must increase in frequency and/or engage additional neurons (*Feldman and Kam, 2015*; *Kam et al., 2013a*). Because burst generation is all-or-none, we propose that burstlets become bursts when a functional threshold of activity is reached (*Feldman and Kam, 2015*; *Kam et al., 2013a*). This threshold mechanism is distinct from the assembly process governing burstlet generation and likely involves activation of persistent inward conductances (*Phillips et al., 2019*; *Picardo et al., 2019*) and recruitment of an additional subpopulation of preBötC neurons that may have a higher threshold for firing (*Kam et al., 2013a*). Inspiratory preBötC somatostatin-expressing neurons, which do not show preinspiratory activity, or a population of premotoneurons that alter burst amplitude when photoablated are candidates for this recruited subpopulation and may, along with Dbx1$^+$ neurons generating both burstlets and bursts, represent an output layer for inspiratory rhythm (*Ashhad and Feldman, 2019*; *Cui et al., 2016*; *Kam et al., 2013a*; *Tan et al., 2010*; *Wang et al., 2014*). By modulating this threshold mechanism, SP and blockade of inhibitory synaptic transmission decreased, whereas Cd$^{2+}$ increased, the fraction of preBötC events that are burstlets, without altering frequency of these events. These neurons may then be targets of additional modulation to alter preBötC burst amplitude. Inhibition may regulate the threshold to recruit additional neurons during the burst and/or modulate membrane potential to augment or prolong activation of inward currents (*Isaacson and Scanziani, 2011*).

In vertebrate central pattern generators (CPGs), whether the timing signal is propagated, how the activity is distributed to muscle groups, and the intensity and duration of activity transmitted to muscle are typically considered pattern generating functions that are distinct from rhythmogenic mechanisms (*Feldman, 1986*; *Feldman and Kam, 2015*; *McCrea and Rybak, 2008*). A basis for separating rhythm and pattern generation is the phenomenon of 'non-resetting' deletions (*Janczewski and Feldman, 2006*; *McCrea and Rybak, 2008*; *Mellen et al., 2003*). Deletions in locomotor CPGs may occur when neurons in the pattern formation layer are inhibited and the rhythmic signal is not transmitted to individual or multiple muscle groups (*McCrea and Rybak, 2008*). For the respiratory CPG, we suggest that transmission of the timing signal from preBötC to multiple muscle groups can fail at the level of the preBötC (via modulation of burst generating mechanisms) to underlie non-resetting deletions (*Feldman and Kam, 2015*). Inspiratory rhythmogenesis and the initial stages of inspiratory pattern generation are therefore mechanistically distinct, but are functions that can be performed in networks that overlap, perhaps significantly (*Feldman and Kam, 2015*; *Kam et al., 2013a*). Thus, burstlet rhythm constitutes the metronomic activity underlying rhythmic breathing movements, and preBötC bursts, by transmitting the burstlet timing signal as activity that has duration, amplitude, and shape to downstream motoneurons, are the first stage of a multilayer pattern formation process. The ultimate pattern of inspiratory activity determined by this microcircuitry includes both the temporal pattern (whether each burstlet beat is transmitted) and the strength and duration of muscle activity (*Feldman and Kam, 2015*; *Janczewski and Feldman, 2006*; *Kam et al., 2013a*; *McCrea and Rybak, 2008*; *Mellen et al., 2003*).

Understanding this organization and the interaction between excitatory neuromodulatory systems with opioids in respiratory neural circuits is relevant for addressing the depressive effects of opioids on breathing. Opioids are the most commonly prescribed drugs for severe acute and chronic pain and play an important role in palliative care. However, with a high potential for addiction, opioid overprescription and abuse has created a pressing public health crisis that affects millions (*Center for Behavioral Health Statistics and Quality, 2018*). Morbidity and mortality from opioid addiction and overdose is largely a result of opioid-induced depression of breathing (*Jaffe and Martin WR, 1990*). Opioids given systemically will act on opioid receptors throughout the nervous system (*Kibaly et al., 2019*), and μORs are expressed in various brain structures regulating breathing (*Gray et al., 1999*; *Manzke et al., 2003*; *Phillips et al., 2012*; *Pokorski and Lahiri, 1981*; *Prkic et al., 2012*; *Zhang et al., 2007*; *Zhang et al., 2011*). While depression of breathing is unlikely to be dependent on actions at a single site (*Lalley et al., 2014*; *Montandon and Horner, 2014*;

*Stucke et al., 2015*), our data are consistent with compelling in vitro and in vivo evidence that the preBötC is the most sensitive site to μOR agonists and that it is largely responsible for respiratory frequency depression by opioids (*Bachmutsky et al., 2019*; *Ballanyi and Ruangkittisakul, 2009*; *Janczewski et al., 2002*; *Montandon and Horner, 2014*; *Montandon et al., 2011*; *Varga et al., 2019*).

In summary, our findings support an essential role for burstlets in respiratory rhythm generation and suggest that opioids act on μORs in preBötC Dbx1[+] neurons to slow burstlet frequency and depress breathing. The depressive effect of the μOR agonist DAMGO on preBötC burstlet frequency, in the absence of a change in burstlet fraction, is consistent with the hypothesis that rhythm and pattern generating mechanisms within the preBötC are separable. We propose that having distinct mechanisms generating burstlets and bursts provides functional substrates for targeted modulation of the rhythmic timing and/or pattern of breathing by other brain areas and neuromodulators. This organization enables the dynamic temporal and pattern lability that is characteristic of breathing and essential for adaptation to changing metabolic demands and coordination with other respiratory-related and orofacial behaviors. Opioids are able to overcome these safety factors by targeting μORs on preBötC neurons responsible for generating rhythm and directly modulating their activity. A better understanding of the mechanisms of μOR actions in respiratory neural circuits has important implications for developing stimulants that excite breathing via non-opioidergic pathways (*van der Schier et al., 2014*) and drugs that reverse opioid actions on preBötC Dbx1[+] neurons to maintain breathing without affecting analgesia.

## Materials and methods

Experimental procedures were carried out in accordance with the United States Public Health Service and Institute for Laboratory Animal Research Guide for the Care and Use of Laboratory Animals. All animals were handled according to approved institutional protocols at the University of California, Los Angeles (#1994-159-83P) and Rosalind Franklin University of Medicine and Science (#B14-16, #B18-10). All protocols were approved by University of California Animal Research Committee (Animal Welfare Assurance #A3196-01) and the Rosalind Franklin University of Medicine and Science Institutional Animal Care and Use Committee (Animal Welfare Assurance #A3279-01). Every effort was made to minimize pain and discomfort, as well as the number of animals.

### Transgenic mice

*Dbx1^{Cre}* mice (*Bielle et al., 2005*) were crossed with either *Rosa26^{tdTomato}* (JAX Stock No. 007908) or *Oprm1^{fl/fl}* (*Weibel et al., 2013*) mouse lines. *Dbx1^{Cre};Rosa26^{tdTomato}* were used to visualize Dbx1[+] neurons, and *Dbx1^{Cre};Oprm1^{fl/fl}* mice were used to delete μORs selectively in Dbx1[+] neurons using cre-lox recombination. Animal genotypes were verified via real time polymerase chain reaction using primers specific for the *Oprm1^{fl/fl}* conditional allele or cre recombinase (*Pierani et al., 2001*; *Rose et al., 2009*; *Weibel et al., 2013*) or by direct visualization of fluorescent reporter. For electrophysiology experiments using these mice, the experimenter was blind to genotype.

### Immunohistochemistry and confocal imaging

*Dbx1^{Cre};Rosa26^{tdTomato}* and *Dbx1^{Cre};Oprm1^{fl/fl}* mice were anesthetized at P0-4 by inhalation of isoflurane. The brainstem was isolated from the pons to the rostral cervical spinal cord, fixed in 4% paraformaldehyde, and cryoprotected in 30% sucrose before being sectioned at 70 mm thickness using CRYOSTAR NX70 (Thermo Scientific). Tissue sections were washed in PBS, and incubated in primary antibody at least overnight, followed by incubation in secondary antibody for 2 hr at room temperature, and cover-slipped in Vectashield with or without DAPI. Primary antibodies used included rabbit anti-μOR (1:400), goat anti-ChAT (1:400; SCBT); secondary antibodies were species specific (1:250) and conjugated to Alexa 488, Rhodamine or Cy5 (Invitrogen or Jackson ImmunoResearch). Sections were imaged using confocal laser scanning microscopy. Confocal images were captured on a Zeiss LSM 710 Meta confocal microscope implemented on an upright Axioplan 2 microscope. Zen software was used to capture images. Confocal images were analyzed with ImageJ, and the final figures were composed in Adobe Photoshop and/or Microsoft PowerPoint.

## In situ hybridization

As described above, mice were anesthetized at P0-4 by inhalation of isoflurane. The isolated brainstem was rapidly frozen in dry ice and stored at −80℃ until sectioning. Transverse sections (14–20 µm) were cut on a cryostat (CryoStar NX70, Thermo Scientific) and mounted on SuperFrost Plus (Fisher Scientific) slides. Slides were processed for fluorescence in situ hybridization of three target RNAs simultaneously according to manufacturer protocols for RNAScope (Advanced Cell Diagnostics). Briefly, fresh frozen tissue samples were postfixed in 10% neutral buffered formalin, washed, and dehydrated in sequential concentrations of ethanol (50, 70% and 100%) at RT. Samples were treated with protease IV, then incubated for 2 hr at 40℃ in the presence of target probes to allow for hybridization. The following target probes were used: Mm-Oprm1, Mm-Tac1, and Mm-ChAT. A series of three amplification steps was necessary to provide substrates for target fluorophore labeling (ChAT conjugated to fluorescein, 1:1200; Oprm1 to Cy5, 1:1200; and Tac1 to Cy3, 1:1500; Jackson ImmunoResearch). After labeling, samples were counterstained with DAPI. ProLong Gold (Invitrogen), was applied to each slide prior to cover slipping. Images were acquired on a confocal laser scanning microscope (LSM710 META, Zeiss). High-resolution z-stack confocal images were taken at 0.3 µm intervals. Cell-by-cell µOR expression profiles were quantified in selected regions of interest (TAC1-positive neurons in preBötC) according to a five-grade scoring system recommended by the manufacturer (ACD score; 0, no staining; 1, 1–3 dots/cell; 2, 4–9 dots/cell; 3, 10–15 dots/cell; 4,>15 dots/cell), where a dot (>0.60 µm) represented a single RNA transcript. A Histo score (H-score) was then calculated from the data, using image-based software analysis (FIJI):

$$H-\mathrm{score} = \sum_{\substack{bin \\ 0-4}}^{n} (\mathrm{ACD\ score\ x\ percentage\ of\ cells\ per\ bin})$$

## Slice preparation and electrophysiology

Neonatal C57BL/6 or mutant mice (P0-5) of either sex were used for cutting brainstem medullary transverse slices (550–600 µm thick) containing the rhythm-generating preBötC, XII inspiratory motoneurons, and premotoneurons linking these populations (*Koizumi et al., 2008*; *Smith et al., 1991*). To obtain slices with the preBötC at the surface, the rostral cut was made above the first set of XII nerve rootlets at the level of the dorsomedial cell column and principal lateral loop of the inferior olive, and the caudal cut captured the obex (*Ruangkittisakul et al., 2011*). Slice cutting artificial cerebrospinal fluid (ACSF) solution was composed of (in mM): 124 NaCl, 3 KCl, 1.5 CaCl$_2$, 1 MgSO$_4$, 25 NaHCO$_3$, 0.5 NaH$_2$PO$_4$, and 30 D-glucose, equilibrated with 95% O$_2$ and 5% CO$_2$, 27℃, pH 7.4. Slices were placed rostral side up in a chamber and perfused at 2–3 ml/min with 28–30℃ recording ACSF. Baseline recording ACSF consisted of either the cutting ACSF solution with K$^+$ raised to 9 mM and Ca$^{2+}$ maintained at 1.5 mM (9/1.5 ACSF) for a robust burst rhythm or the cutting ACSF solution with K$^+$ maintained at 3 mM and Ca$^{2+}$ lowered to 1 mM (3/1 ACSF) to study burstlets and bursts together (*Kam et al., 2013a*). Drugs, including DAMGO, SP, picrotoxin, strychnine, and CdCl$_2$, were obtained from Sigma-Aldrich and bath-applied at the specified concentrations. The concentrations for each drug were chosen based on their demonstrated efficacy in in vitro preparations and well-below levels that would produce off-target receptor binding as shown either by dose-response curves or ineffectiveness in receptor knockout mice (*Gray et al., 1999*; *Ptak et al., 2000*; *Ren and Greer, 2006*). In all experiments, slices were allowed to equilibrate for 30 min to ensure that the frequency and magnitude of XII and preBötC population activities reached steady-state. Respiratory activity reflecting suprathreshold action potential firing from populations of neurons was recorded from XII motor nucleus or nerve roots and as population activity directly from the preBötC using suction electrodes (tip size ~50 µm) and an Axiopatch 200A (Molecular Devices), Multiclamp 700B (Molecular Devices), and/or a differential AC amplifier (AM systems), filtered at 2–4 kHz, integrated, and digitized at 10 kHz. Integration was performed on custom built Paynter filter with a 20–100 ms time constant. Digitized data were analyzed off-line using custom procedures written for IgorPro (Wavemetrics).

## Holographic photostimulation

Holographic photostimulation was performed on a Phasor SLM system (Intelligent Imaging Innovations, Inc) mounted around an epifluorescence upright microscope (Axioscope; Zeiss). We used a 405 nm diode laser (CUBE 405–100; Coherent) to uncage MNI-glutamate (0.5 mM) and depolarize targeted neurons. An iterative Fourier transform algorithm, implemented in Slidebook 5 (Intelligent Imaging Innovations, Inc), computed the phase pattern on the SLM corresponding to the desired distribution of light intensity at the focal plane of the objective. A blocker was inserted in the intermediate Fourier plane to block the unmodulated light component (zero order spot) and replicate patterns (*Golan et al., 2009*). Neurons were targeted by centering 10 μm circular spots over the somata of the selected neurons. The laser intensity per spot was set at 1–3 mW, which was previously determined to elicit bursts of action potentials in the targeted neurons without the spread of glutamate to neighboring neurons (*Kam et al., 2013b*). Each laser stimulation consisted of $5 \times 0.8$ ms pulses, delivered at 200 Hz. A threshold stimulus was determined as the minimum number of neurons and minimum laser power required to entrain preBötC bursts at slightly faster than the endogenous frequency with a success rate >80%. The latency between laser stimulation and burst initiation was calculated as the time between the start of the next preBötC burst and the beginning of laser stimulation. preBötC activity was used here, rather than XII activity, to isolate the effects of DAMGO on rhythmogenic mechanisms and avoid additional delays or failures due to depression of XII premotoneurons and motoneurons. Successful triggering of a burst by photostimulation was defined as burst detection within 0.5 s of laser stimulation.

## Data analysis and statistics

Semiautomated event detection of respiratory-related activity recorded in XII output or preBötC population recordings was performed using custom procedures written in IgorPro. Multiple criteria, including slope and amplitude thresholds, were used to select events automatically, which were then confirmed visually. Event duration, amplitude, shape, and synchrony between XII and preBötC activity were criteria used to categorize detected events as burstlets, bursts, and doublets. Burstlets were events in preBötC that did not temporally overlap a XII burst. Doublets were distinguished from bursts by their longer duration and the presence of multiple peaks of activity. Two closely spaced bursts were considered a doublet based on the distribution of the period of XII output. A small peak at <2 s in the distribution of periods of XII bursts was usually observed. A Gaussian was fit to this small peak and a threshold time interval was set. Two bursts separated by less than this threshold were considered a doublet. As doublets were in phase with bursts and burstlets, these events were grouped with bursts and not separately analyzed.

Rhythmogenic processes were captured by measuring the frequency of preBötC rhythmic activity, which included small amplitude burstlets that were not transmitted to XII motor output, larger amplitude bursts that produced XII bursts, and longer duration doublets that were also transmitted to XII motoneurons. The average frequency was calculated as the mean of the instantaneous frequencies (1/interevent interval) across all preBötC activity (burstlets, bursts, and doublets) in that condition. Pattern generating processes that included the production and transmission of preBötC activity to XII output were characterized by measuring the fraction of preBötC events that were burstlets. The burstlet fraction was calculated as the ratio of the number of burstlets to the total number of rhythmic preBötC events (burstlets, bursts, and doublets). Amplitudes were normalized to the amplitude of the largest burst in the control condition.

Data are expressed as mean ± standard deviation. In figures displaying average data, data points from individual experiments are displayed in gray. To determine sample sizes, a priori power analysis was performed using G*power 3.1.9.2 (*Faul et al., 2009*). In this program, the appropriate statistical test along with a desired power of 95% at a 5% significance level was selected for each type of experiment with effects sizes based on published or preliminary data. These parameters yielded minimum total sample sizes of 6 for all tests. Sample sizes represent the number of biological replicates for each group. Statistical analysis was performed using IgorPro or GraphPad (PRISM software). Comparisons between two groups were performed with paired or unpaired Student's *t*-tests when normality assumptions were met, and Mann-Whitney *U*-tests in non-parametric cases. Multiple comparisons were evaluated using parametric one-way or two-way repeated measures ANOVA. Post-hoc Tukey or paired Student's *t*-tests with the Bonferroni-Holm comparison were then used to

determine significance for within group comparisons. Outliers were defined as data points greater than three standard deviations from the mean. To test for correlations between burstlet fraction and preBötC $f$, we determined a best-fit line and reported the coefficient of determination ($r^2$), a measure of goodness of fit. We applied an $F$-test to evaluate the null hypothesis that the slope of the fitted line was not different from zero, that is there was no relationship between burstlet fraction and preBötC $f$. To determine whether the slope of the linear model changed across conditions, that is whether drug application changed the correlation, we performed an analysis of covariance (ANCOVA) and applied a $t$-test for an ANCOVA with two groups or an $F$-test for an ANCOVA with more than two groups that tests for the equality of the slopes of the linear model across conditions (homogeneity of regression slopes). Statistical significance was set at $p<0.05$.

## Acknowledgements

The authors thank Grace Li for excellent technical work. $Oprm1^{fl/fl}$ mice were generously provided by Dr. Wendy Weibel, Dr. Chris Evans, and the Animal Breeding Core at UCLA (NIDA P50 Center Grant, DA005010). This work was supported by National Institutes of Health grants NS072211, HL135779, and NS097492; and an international postdoctoral grant from the Swedish Research Council (Vetenskapsrådet).

## Additional information

### Funding

| Funder | Grant reference number | Author |
| --- | --- | --- |
| National Institutes of Health | NS072211 | Jack L Feldman |
| National Institutes of Health | HL135779 | Jack L Feldman |
| National Institutes of Health | NS097492 | Kaiwen Kam |
| Vetenskapsrådet | | Carolina Thörn Pérez |

The funders had no role in study design, data collection and interpretation, or the decision to submit the work for publication.

### Author contributions

Xiaolu Sun, Carolina Thörn Pérez, Conceptualization, Formal analysis, Investigation, Visualization; Nagaraj Halemani D, Xuesi M Shao, Morgan Greenwood, Sarah Heath, Formal analysis, Investigation; Jack L Feldman, Conceptualization, Supervision, Funding acquisition, Project administration; Kaiwen Kam, Conceptualization, Resources, Data curation, Formal analysis, Supervision, Funding acquisition, Investigation, Visualization, Methodology, Project administration

### Author ORCIDs

Carolina Thörn Pérez (iD) https://orcid.org/0000-0002-3480-8599
Xuesi M Shao (iD) https://orcid.org/0000-0002-5165-347X
Jack L Feldman (iD) https://orcid.org/0000-0003-3692-9412
Kaiwen Kam (iD) https://orcid.org/0000-0002-8479-0542

### Ethics

Animal experimentation: Experimental procedures were carried out in accordance with the United States Public Health Service and Institute for Laboratory Animal Research Guide for the Care and Use of Laboratory Animals. All of the animals were handled according to approved institutional protocols at the University of California, Los Angeles (#1994-159-83P) and Rosalind Franklin University of Medicine and Science (#B14-16, #B18-10). All protocols were approved by University of California Animal Research Committee (Animal Welfare Assurance #A3196-01) and the Rosalind Franklin University of Medicine and Science Institutional Animal Care and Use Committee (Animal Welfare Assurance #A3279-01). Every effort was made to minimize pain and discomfort, as well as the number of animals.

Decision letter and Author response

Decision letter https://doi.org/10.7554/eLife.50613.sa1

Author response https://doi.org/10.7554/eLife.50613.sa2

## Additional files

### Supplementary files

- Supplementary file 1. Correlation of burst and burstlet parameters and preBötC $f$.

- Source data 1. Source data for graphs.

- Transparent reporting form

### Data availability

All data generated or analysed during this study are included in the manuscript and supporting files. Source data files are available with the paper and also at https://sites.google.com/site/kwkamlab/source_data.

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
