## [Decision Letter]

**Acceptance summary:**

This paper addresses a persistent problem in understanding how rhythm emerges from the neuronal circuit that controls the inspiratory cycle of breathing in mammals within the preBötC of the brainstem. Understanding how rhythm arises here has implications for other mammalian central pattern generating networks that also rely on interconnected glutamatergic interneurons. Recent work by this group has implicated a process within a glutamatergic subpopulation of Dbx1-derived (Dbx1^+^) interneurons of preBötC that are spontaneously spiking and organize into synchronous activation that is manifest as a burstlet. These burstlets can be sub-threshold in some way for premotor burst generation. In medullary slices from neonatal mice, they find that the μ-opioid receptor (μOR) agonist DAMGO slowed burstlet generation. Opioid-mediated depression was abolished by genetic deletion of μORs in a glutamatergic preBötC subpopulation and was reduced by the neuropeptide Substance P, but not blockade of inhibitory synaptic transmission. They conclude that opioids alter a rhythmogenic process giving rise to burstlets that does not rely on strong bursts of activity associated with motor output.

**Decision letter after peer review:**

Thank you for submitting your article "Opioids modulate an emergent rhythmogenic process to depress breathing" for consideration by *eLife*. Your article has been reviewed by three peer reviewers, and the evaluation has been overseen Ronald Calabrese as the Senior and Reviewing Editor. The following individual involved in review of your submission has agreed to reveal their identity: Shawn Hochman (Reviewer #3).

The reviewers have discussed the reviews with one another and the Reviewing Editor has drafted this decision to help you prepare a revised submission.

Essential revisions:

The reviewers were in general enthusiastic about the data but had some significant concerns about how the data is presented and interpreted. The individual reviewer comments should all be addressed. To help focus the revision, we offer a few consensus comments.

1) The results with SP seem to counter the main hypothesis, and the alternative explanation of a ceiling effect is not fully convincing.

2) The reviewers did not think that framing the work in terms of rhythm vs. pattern generation was useful and might be counterproductive to getting at the real interest in the work, which is how burst activity begins and how 'full' premotor network activation is achieved. Similarly the discussion of emergence as part of the rhythm generating mechanism was not wholly convincing and needs specific clarification.

3) Presentation of the data should be clarified and reviewer suggested new analyses added.

Reviewer #1:

1) The notion of rhythm vs. pattern has little heuristic value in the preBötC; there is no pattern here just the presence or absence of a full premotor burst. In the full respiratory CPG and in other mammalian CPGs like that for locomotion, rhythm vs. pattern makes sense; muddying the waters with this usage here may become quite counterproductive to getting at the mechanism that the authors are interested in. Are the authors suggesting that a different subpopulation of glutamatergic Dbx1-derived (Dbx1^+^) interneurons drives down stream motor neurons than those that participate in burstlet formation? If so, please be explicit.

2) The vagueness of the rhythmogenic process as presented in the Discussion leave this reader confused. What exactly do the authors imagine is happening in this network? Can you be more explicit in your hypothesis? What is emergent? What is the threshold you envision that converts a burstlet to a burst? The Discussion states "We suggest that DAMGO-mediated hyperpolarization and/or decreases in synaptic probability of release in and among μOR-expressing preBötC Dbx1^+^ neurons reduce spontaneous firing rate, weaken the propagation of action potentials to postsynaptic neurons, and increase the number of inputs required to reach action potential threshold. These effects reduce the moment-to-moment probability that a critical number of neurons will be simultaneously active and prolong the time required for preBötC neurons to increase their firing, achieve synchrony, and generate burstlets (Ashhad and Feldman, 2019; Feldman and Kam, 2015; Kam et al., 2013a). The pattern-generating mechanism that converts burstlets to bursts may, by analogy, require a higher threshold of activity and synchrony be reached via independently regulated, distinct mechanisms, such as the activation of persistent inward conductances (Picardo et al., 2019)." There are many things here that are vague. Please be explicit about how action potentials are propagated through the system to what postsynaptic neurons and how action potential threshold affects this process. What generates the activity in the first place? Do you envision some spontaneously active bursters or tonic firers? How is the "critical number of neurons" related to the less than ten randomly selected preBötC Dbx1^+^ neurons that can induce an extopic burst? Why is the mechanism that converts burstlets to bursts called a pattern generating mechanism? Is this some kind of threshold? Is the threshold determined by the number of preBötC Dbx1^+^ neurons that are active, thus forming some kind of bigger burstlet? Must the neurons participating in a burstlet be μOR-expressing? The authors need not answer all these questions in rebuttal, but they should obviate a lot of the confusion that gives rise to them by being more explicit.

Reviewer #2:

In this manuscript Sun et al. use a set of complementary experimental strategies to test the hypothesis that inspiratory rhythmogenesis manifested as burstlets is an emergent process by examining how opioid modulation affects the frequency of these burstlets. The results reported here support this hypothesis; they show that the µ-opioid receptor agonist DAMGO decreases the burstlet frequency and the respiratory motor output. These effects persisted after blockade of inhibitory synaptic transmission, but were reduced by the neuropeptide substance P. The µ-opioid receptors were found in a glutamatergic subpopulation in the preBötzinger complex (preBötC) and restricted genetic deletion of these receptors in these neurons abolished the effects of DAMGO. This is a clear study reporting strong support for the hypothesis that burstlets represent an emergent process underlying rhythmogenesis in the preBötC. I have some points that could help the clarity of the presentation this manuscript.

1) It would be helpful to provide additional explanations about the rhythm and pattern generation. The burstlets and inspiratory bursts are interlinked and it is not clear why the inspiratory burst would only be generated by pattern- and not rhythm-related mechanisms. It is also not clear how the lack of a change in the burstlet fraction is consistent with the hypothesis that the rhythm and pattern generation mechanisms are separable.

2) It should be made clear in the Results section that photostimulation was used to uncage glutamate. Figure 2: Could you produce burstlets using photostimulation? And could you counteract the DAMGO-induced depression of the burst by increasing the number of the activated neurons?

3) The rescue of DAMGO-induced depression by SP in vitro is interesting, have the authors tried to rescue the opioid-induced depression of respiration in vivo? Such results would further enhance the impact of this study.

4) Figure 3: The panels and the text are too small to clearly distinguish the positive and negative neurons. The authors should provide quantification of the number of positive neurons and number of puncta. As it stands, it is very difficult to evaluate the difference between samples from animals with and without µ-opioid receptors.

Reviewer #3:

The present manuscript tests the hypothesis that small subthreshold burstlet activity reflects an emergent rhythmogenic process that coalesce into bursts associated with respiratory inspiration.

The authors use an in vitro model and transgenic approaches to demonstrate that the μOR expressed in preBotC (Dbx1) glutamatergic neurons essential to inspiratory rhythmogenesis are responsible for μOR mediated (DAMGO) respiratory depression.

Observed threshold bursts relative to the total number of subthreshold burstlets were used to support the hypothesis that depressive effects were associated with corresponding reduced incidence of burstlets, and thereby identifying a subthreshold rhythm-generating neural-network mechanism of action.

The work makes an important contribution to understanding the role of μORs in the depressant actions on respiratory rhythmogenesis.

Overall, experimental design is strong and methods employed are impressive and elegant. The authors make strategic use of the Dbx1Cre line to demonstrate its essential locus for μOR respiratory depressive actions by reducing incidence of burstlets necessary to coalesce into bursts.

There are several areas in the Results section that could be clarified by small adjustments in data presentation. The following structural change to some panels would be particularly helpful.

1) Given the emphasis on the importance of burstlets in this paper, it would have been most helpful to compare relations in terms of burstlet fraction and preBotC f in all panels where this is relevant (particularly Figure 4C and Figure 4—figure supplement 1 should have equivalent panel presentation as seen in Figure 4E and G).

---

## [Author Response]

Essential revisions:The reviewers were in general enthusiastic about the data but had some significant concerns about how the data is presented and interpreted. The individual reviewer comments should all be addressed. To help focus the revision, we offer a few consensus comments.1) The results with SP seem to counter the main hypothesis, and the alternative explanation of a ceiling effect is not fully convincing.

We understand the confusion caused by inclusion of some of the SP data, particularly the SP effect in the Dbx1^cre^;Oprm1^fl/fl^ mouse and agree with the reviewers that these data, while interesting and worthy of further pursuit, may be more appropriate for a separate study. We have removed the supplementary figure and revised the Results and Discussion to focus our manuscript. We also appreciate some of the alternative interpretations provided by the reviewers, and think that SP effects on neurons outside the preBötC that modulate inspiratory rhythm and pattern may be responsible for the state-dependent effects we observe. Again, we agree these inquiries may be appropriate for a separate study.

2) The reviewers did not think that framing the work in terms of rhythm vs. pattern generation was useful and might be counterproductive to getting at the real interest in the work, which is how burst activity begins and how 'full' premotor network activation is achieved. Similarly the discussion of emergence as part of the rhythm generating mechanism was not wholly convincing and needs specific clarification.

We appreciate the reviewers for recognizing the significance of the manuscript as it relates to burstlets and mechanisms of inspiratory rhythmogenesis. We acknowledge that our framing may not have been sufficiently clear. We have largely removed references to pattern generation from the Introduction and Results and lay out our conception of burstlets and bursts as they relate to rhythm and pattern generation in the Discussion. We elaborate on the emergent aspects of our model more fully in both the Introduction and Discussion.

3) Presentation of the data should be clarified and reviewer suggested new analyses added.

All suggested changes were either made or addressed below, and new analysis was added.

Reviewer #1:1) The notion of rhythm vs. pattern has little heuristic value in the preBötC; there is no pattern here just the presence or absence of a full premotor burst. In the full respiratory CPG and in other mammalian CPGs like that for locomotion, rhythm vs. pattern makes sense; muddying the waters with this usage here may become quite counterproductive to getting at the mechanism that the authors are interested in. Are the authors suggesting that a different subpopulation of glutamatergic Dbx1-derived (Dbx1^+^) interneurons drives down stream motor neurons than those that participate in burstlet formation? If so, please be explicit.

See response to consensus comment #2 above. Also, there is indeed evidence that a subpopulation of Dbx1^+^ neurons, which primarily fires during inspiration and does not show preinspiratory activity, constitutes part of an output layer (SST^+^). The novel aspect of our model, which may be unique to breathing, is that the first patterning layer includes these neurons as well as Dbx1^+^ neurons that generate burstlets and bursts. So pattern formation begins as a distinct process from rhythmogenesis that manifests as bursting in a preBötC population that overlaps with rhythmogenic neurons. We consider bursting as the initial stage of pattern generation as it is the burst that is transmitted to and further shaped by downstream premotor and motoneurons. Furthermore, we consider bursts critical for determining the temporal pattern, i.e., whether the burstlet beat generates motor activity. We now describe this explicitly in the Discussion.

2) The vagueness of the rhythmogenic process as presented in the Discussion leave this reader confused. What exactly do the authors imagine is happening in this network? Can you be more explicit in your hypothesis? What is emergent? What is the threshold you envision that converts a burstlet to a burst? The Discussion states "We suggest that DAMGO-mediated hyperpolarization and/or decreases in synaptic probability of release in and among μOR-expressing preBötC Dbx1^+^ neurons reduce spontaneous firing rate, weaken the propagation of action potentials to postsynaptic neurons, and increase the number of inputs required to reach action potential threshold. These effects reduce the moment-to-moment probability that a critical number of neurons will be simultaneously active and prolong the time required for preBötC neurons to increase their firing, achieve synchrony, and generate burstlets (Ashhad and Feldman, 2019; Feldman and Kam, 2015; Kam et al., 2013a). The pattern-generating mechanism that converts burstlets to bursts may, by analogy, require a higher threshold of activity and synchrony be reached via independently regulated, distinct mechanisms, such as the activation of persistent inward conductances (Picardo et al., 2019)." There are many things here that are vague. Please be explicit about how action potentials are propagated through the system to what postsynaptic neurons and how action potential threshold affects this process. What generates the activity in the first place? Do you envision some spontaneously active bursters or tonic firers? How is the "critical number of neurons" related to the less than ten randomly selected preBötC Dbx1^+^ neurons that can induce an extopic burst? Why is the mechanism that converts burstlets to bursts called a pattern generating mechanism? Is this some kind of threshold? Is the threshold determined by the number of preBötC Dbx1^+^ neurons that are active, thus forming some kind of bigger burstlet? Must the neurons participating in a burstlet be μOR-expressing? The authors need not answer all these questions in rebuttal, but they should obviate a lot of the confusion that gives rise to them by being more explicit.

We made substantial revisions to the Discussion to be more explicit about the proposed mechanism.

Reviewer #2:In this manuscript Sun et al. use a set of complementary experimental strategies to test the hypothesis that inspiratory rhythmogenesis manifested as burstlets is an emergent process by examining how opioid modulation affects the frequency of these burstlets. The results reported here support this hypothesis; they show that the µ-opioid receptor agonist DAMGO decreases the burstlet frequency and the respiratory motor output. These effects persisted after blockade of inhibitory synaptic transmission, but were reduced by the neuropeptide substance P. The µ-opioid receptors were found in a glutamatergic subpopulation in the preBötzinger complex (preBötC) and restricted genetic deletion of these receptors in these neurons abolished the effects of DAMGO. This is a clear study reporting strong support for the hypothesis that burstlets represent an emergent process underlying rhythmogenesis in the preBötC. I have some points that could help the clarity of the presentation this manuscript.1) It would be helpful to provide additional explanations about the rhythm and pattern generation. The burstlets and inspiratory bursts are interlinked and it is not clear why the inspiratory burst would only be generated by pattern- and not rhythm-related mechanisms. It is also not clear how the lack of a change in the burstlet fraction is consistent with the hypothesis that the rhythm and pattern generation mechanisms are separable.

We elaborate about the proposed mechanism in the Introduction and Discussion. See response to consensus comment #2 above. Also, the effects of Cd^2+,^ which selectively delete bursts, as well as the various manipulations that specifically alter either preBötC ƒ or burstlet fraction strongly suggest that the mechanisms generating rhythm and burstlets can be separated from burst generation. The primary link between the two is that these mechanisms both involve a population of preBötC neurons.

2) It should be made clear in the Results section that photostimulation was used to uncage glutamate. Figure 2: Could you produce burstlets using photostimulation? And could you counteract the DAMGO-induced depression of the burst by increasing the number of the activated neurons?

We clarified the use of MNI-glutamate in the Results, figure legend, and Materials and methods. The question of whether we could produce burstlets with photostimulation is a very interesting one, and we are pursuing that question. We would predict that increasing the number of activating neurons might counteract DAMGO-induced depression; however, we do not know how many more might be required and a negative result would be difficult to interpret. Such an experiment would potentially require examining a large parameter range and is beyond the scope of this study.

3) The rescue of DAMGO-induced depression by SP in vitro is interesting, have the authors tried to rescue the opioid-induced depression of respiration in vivo? Such results would further enhance the impact of this study.

We agree that testing whether SP could counteract opioid-induced respiratory depression in vivowould be highly significant; however, we feel that such experiments are outside the scope of a study focused primarily on testing the burstlet hypothesis with opioids. We suggest that our results here could certainly inform such therapies.

4) Figure 3: The panels and the text are too small to clearly distinguish the positive and negative neurons. The authors should provide quantification of the number of positive neurons and number of puncta. As it stands, it is very difficult to evaluate the difference between samples from animals with and without µ-opioid receptors.

We enlarged the figure. The H-scores for the two groups quantify the level of µOR mRNA expression and were described in the text. For the supplementary figure, we provide new figures and quantification for the number of TAC1/µOR/NK1R co-expressing neurons.

Reviewer #3:The present manuscript tests the hypothesis that small subthreshold burstlet activity reflects an emergent rhythmogenic process that coalesce into bursts associated with respiratory inspiration.The authors use an in vitro model and transgenic approaches to demonstrate that the μOR expressed in preBotC (Dbx1) glutamatergic neurons essential to inspiratory rhythmogenesis are responsible for μOR mediated (DAMGO) respiratory depression.Observed threshold bursts relative to the total number of subthreshold burstlets were used to support the hypothesis that depressive effects were associated with corresponding reduced incidence of burstlets, and thereby identifying a subthreshold rhythm-generating neural-network mechanism of action.The work makes an important contribution to understanding the role of μORs in the depressant actions on respiratory rhythmogenesis.Overall, experimental design is strong and methods employed are impressive and elegant. The authors make strategic use of the Dbx1Cre line to demonstrate its essential locus for μOR respiratory depressive actions by reducing incidence of burstlets necessary to coalesce into bursts.There are several areas in the Results section that could be clarified by small adjustments in data presentation. The following structural change to some panels would be particularly helpful.1) Given the emphasis on the importance of burstlets in this paper, it would have been most helpful to compare relations in terms of burstlet fraction and preBotC f in all panels where this is relevant (particularly Figure 4C and Figure 4—figure supplement 1 should have equivalent panel presentation as seen in Figure 4E and G).

We performed the analysis and added the data to the figure (now Figure 4B). We removed the original Figure 4—figure supplement 1.